# MD Simulations to Calculate NMR Relaxation Parameters of Vanadium(IV) Complexes: A Promising Diagnostic Tool for Cancer and Alzheimer’s Disease

**DOI:** 10.3390/ph16121653

**Published:** 2023-11-27

**Authors:** Rodrigo Mancini Santos, Camila Assis Tavares, Taináh Martins Resende Santos, Hassan Rasouli, Teodorico Castro Ramalho

**Affiliations:** 1Laboratory of Molecular Modelling, Department of Chemistry, Federal University of Lavras, Lavras 37200-000, MG, Brazil; rodrigomancini4@gmail.com (R.M.S.); tainah-martins@hotmail.com (T.M.R.S.); h3n.rasouli@gmail.com (H.R.); 2Medical Biology Research Center (MBRC), Kermanshah University of Medical Sciences, Kermanshah 6714414971, Iran; 3Department of Chemistry, Faculty of Science, University of Hradec Králové, 500 03 Hradec Králové, Czech Republic

**Keywords:** NMR relaxation, computational chemistry, molecular dynamics, OWSCA, vanadium complexes

## Abstract

Early phase diagnosis of human diseases has still been a challenge in the medicinal field, and one of the efficient non-invasive techniques that is vastly used for this purpose is magnetic resonance imaging (MRI). MRI is able to detect a wide range of diseases and conditions, including nervous system disorders and cancer, and uses the principles of NMR relaxation to generate detailed internal images of the body. For such investigation, different metal complexes have been studied as potential MRI contrast agents. With this in mind, this work aims to investigate two systems containing the vanadium complexes [VO(metf)_2_]·H_2_O (VC1) and [VO(bpy)_2_Cl]^+^ (VC2), being metformin and bipyridine ligands of the respective complexes, with the biological targets AMPK and ULK1. These biomolecules are involved in the progression of Alzheimer’s disease and triple-negative breast cancer, respectively, and may act as promising spectroscopic probes for detection of these diseases. To initially evaluate the behavior of the studied ligands within the aforementioned protein active sites and aqueous environment, four classical molecular dynamics (MD) simulations including VC1 + H_2_O (1), VC2 + H_2_O (2), VC1 + AMPK + H_2_O (3), and VC2 + ULK1 + H_2_O (4) were performed. From this, it was obtained that for both systems containing VCs and water only, the theoretical calculations implied a higher efficiency when compared with *DOTAREM*, a famous commercially available contrast agent for MRI. This result is maintained when evaluating the system containing VC1 + AMPK + H_2_O. Nevertheless, for the system VC2 + ULK1 + H_2_O, there was observed a decrease in the vanadium complex efficiency due to the presence of a relevant steric hindrance. Despite that, due to the nature of the interaction between VC2 and ULK1, and the nature of its ligands, the study gives an insight that some modifications on VC2 structure might improve its efficiency as an MRI probe.

## 1. Introduction

The growing body of evidence suggests that the prevalence of cancer and Alzheimer’s disease (AD) has dramatically increased over the past few decades. It is estimated that the number of worldwide deaths by these diseases has significantly increased yearly [1,2]. Various genetic and environmental risk factors are involved in the progression of cancer and AD; however, despite commonality of risk factors in cancer and AD, studies have shown that there is an inverse correlation between the pathogenesis of AD and progression of cancer [3,4,5]. To date, various FDA-approved drugs have been used to treat both cancer and AD; however, the outcomes suggest that the current therapies could not completely prevent the onset of these diseases and the applied drugs mainly alleviated specific complications associated with cancer and AD.

Early diagnosis is extremely crucial in the battle against AD and cancer, representing a critical turning point in the pursuit of effective treatments and improved patient outcomes. Identifying these diseases in their initial stages empowers individuals and healthcare providers with valuable opportunities for timely intervention, personalized therapies, and enhanced quality of life [6,7]. However, early diagnosis in both AD and cancer is riddled with multifaceted challenges, mostly because of the complexity of molecular pathways and biomolecules involved in the pathogenesis of both [8,9,10,11,12].

Despite these difficulties, MRI is a technique that presents several advantages in the detection of such diseases as AD and cancer and has been vastly used for this purpose [13,14]. It enables non-invasive examination, eliminating the need for invasive procedures or exposure to ionizing radiation, making it a safer option for patients of all ages [15]. The versatility of MRI allows for imaging various tissues, organs, and systems within the body, facilitating the identification and characterization of abnormalities with remarkable precision [16].

Moreover, to enhance the diagnostic potential of MRI even further, researchers have been exploring the use of metal complexes as probes due to their unique magnetic properties [17,18]. Gadolinium-based complexes, for instance, are widely used in clinical practice, exhibiting distinct relaxation properties and potential for functional imaging [19]. Another important fact about gadolinium is that it has potential toxicity and long-term accumulation in the body [20]. Developing new contrast agents that are biocompatible and less prone to accumulation is crucial to ensure patient safety and enhance the overall efficacy and reliability of medical imaging techniques [21].

In the quest for alternative probes in MRI, several potential candidates have emerged that offer improved safety profiles and imaging capabilities [21]. Among those, vanadium(IV) complexes (VCs) have gained significant attention as contrast agents in medical imaging, once they are paramagnetic and due to their unique properties and versatile applications, presenting biocompatibility and favorable magnetic and optical properties [22,23,24].

It is known that the average presence of vanadium in the human body is about 1 mg, being non-toxic at this concentration [25]. However, VCs can exhibit varying levels of toxicity depending on several factors such as the form of vanadate, oxidation state, time of exposure, dose, and route of intoxication [26]. Moreover, the toxicity of vanadium is influenced by the route of administration, species tested, and the specific complexation of the metal [27]. Based on the foregoing evidence, studies considering the use of vanadium for the treatment of diseases should take into account different factors in order to better understand the effects on a biological system. Nevertheless, vanadium complexes have shown significant potential for the treatment of AD, making them intriguing candidates for anticancer and anti-diabetic therapy as well [28,29,30,31].

In addition to being potential targets for the treatment of AD, cancer, and diabetes, VCs show a large spectrum of potential activity and are efficiently used in the treatment of leishmaniasis, Chagas disease, influenza, and dengue [32,33,34]. Furthermore, recent studies have been conducted to investigate the use of VCs as promising candidates for the treatment of COVID-19 due to their antiviral, anti-inflammatory, and hypoglycemic effects [35,36,37]. Recently, VCs such as [VO(metf)_2_]·H_2_O (VC1) and [VO(bpy)_2_Cl]^+^ (VC2), where metf is metformin and bpy is bipyridine, have shown great potential regarding their application to AD and autophagy associated with triple-negative breast cancer, respectively [38,39].

In the study conducted by Tavares et al. [38], the biological application of VC1 in the system containing AMP-activated protein kinase (AMPK) was investigated. AMPK plays a pivotal role in AD and is considered a link between AD and diabetes mellitus type 2. The results obtained by VC1 are encouraging, and the use of such a complex should be taken into consideration in future studies. Whereas in the work reported by Santos, T. M. R. et al. [39], the interaction of VC2 and Unc-51-like kinase 1 (ULK1), inducing autophagy in triple-negative breast cancer cells, was explored. Once again, results showed the versatility and great potential of vanadium complexes.

Considering computational methods, MD simulations present to be a powerful tool to study complex systems, being able to compute solvent effects [40] and consider the presence of biological targets, such as AMPK and ULK1. Moreover, allying MD simulation with less expensive quantum methods (QM) can be very useful to reduce the computational demand and assess NMR properties, which is not computationally feasible using robust QM calculation [40,41,42]. Therefore, in order to ensure correct results and a feasible calculation, previous works [38,39] parametrized a force field for VC1 and VC2, where relevant relativistic effects were also taken into account. Thus, with the presented background, MD simulations could be carried out.

In light of these findings, this work aims to investigate promising VCs as candidates to replace gadolinium as contrast agents for the diagnosis of Alzheimer’s disease and triple-negative breast cancer in order to enhance safety of the MRI technique and reposition its application.

For such purposes, MD simulations were carried out for four systems, being VC1 + WAT (1), VC2 + WAT (2), VC1 + AMPK + WAT (3), and VC2 + ULK1 + WAT (4), where WAT denotes water. Therefore, conformational selection was performed by applying the Optimized Wavelet Signal Compression Algorithm (OWSCA), reducing data without losing important information of the original data. At the end, NMR parameters were estimated from MD simulation data based on calculations presented by Chen, P. et al. and Villa, A. and Stock, G. [43,44].

Finally, the originality of the work relies on the use of VCs as spectroscopic probes for disease detection, repositioning its application with respect to previous mentions in the literature, as well as providing more information on biological probes. It is expected that the presented work may encourage future research, contributing with new insights aiming at VC-based drug design, including the possibility of using VCs for many types of cancer and AD theranostics. Moreover, the use of the OWSCA algorithm as part of the strategy to obtain relaxation parameters offers a unique contribution and a fresh perspective.

## 2. Results and Discussion

### 2.1. Calculating T_i_ and R_i_ from MD Simulation Data

In order to obtain T_i_ and R_i_ values for VC1 and VC2, MD simulation data were used to perform the calculations. As proposed by Chen, P. et al. and Villa, A. and Stock, G. [43,44], the NMR spin relaxation parameters can be obtained from MD simulation data. To perform such calculations, the work published by Gonçalves, M. A. et al. [45] shed some light on the procedure. First, the distances V⋯^1^H were measured in MD trajectory for each conformation selected by the OWSCA methodology, then the autocorrelations of the measured set of distances were computed and fitted. If it is assumed that the overall internal motion of molecules is independent, then the total correlation function C(t), expressed in Equation (1), can be divided into two different equations, the overall motion correlation function C_O_(t) and the internal motion correlation function C_I_(t), expressed by Equations (2) and (3), respectively.
(1)$$C\left(t\right)=C_{O}(t)C_{I}(t)$$
(2)$$C_{O}\left(t\right)=\frac{1}{5}e^{-t/\tau_{c}}$$
(3)$$C_{I}\left(t\right)=S^{2}+(1-S^{2})e^{-t/\tau_{e}}$$
where τ_c_ is the rotational correlation time, S^2^ is the order parameter, and τ_e_ is the effective correlation time. The three mentioned terms are estimated from the above equations, and once this is done, the spectral density J(ω) can be given by the Fourier transform of C(t), resulting in Equation (4).
(4)$$J(\omega )=\frac{2}{5}\left(\frac{S^{2}\tau_{c}}{1+\tau_{c}^{2}\omega^{2}}\frac{{(1-S}^{2})\tau }{1+\tau^{2}\omega^{2}}\right)$$

Being τ^−1^ = τ_c_^−1^ + τ_e_^−1^. This equation can be used to determine the longitudinal and transverse relaxation rates and times, according to Equations (5) and (6), respectively.
(5)$$R_{1}=\frac{1}{T_{1}}=K\left[J\left(\omega_{0}\right)+J(2\omega_{0})\right]$$
(6)$$R_{2}=\frac{1}{T_{2}}=\frac{K}{4}\left[J(0)+10J\left(\omega_{0}\right)+J(2\omega_{0})\right]$$
where K is expressed by Equation (7) and ω_0_ = γ·B_0_, which represents the rate of precession of the magnetic moment of the proton with an external magnetic field B_0_ and gyromagnetic radius of hydrogen γ.
(7)$$K={\left(\frac{\mu_{0}}{4\pi }\right)}^{2}\frac{3}{2}\frac{h^{2}{ϒ}^{4}I(I+1)}{\phi^{6}}$$

With µ_0_ being the vacuum permeability, h being the Planck constant, I being spin quantum number, ϕ being average V⋯^1^H distance, as shown in Figure 1, and γ being the gyromagnetic radius of hydrogen. In this work, B_0_ was considered equal to 1.5 T and γ equal to 42.58 MHz/T. This choice was made because when discussing contrast agents for clinical applications, it is customary to reference R_i_ and T_i_ values at 1.5 T and a standard body temperature of 37 °C. [46].

### 2.2. Validation of the T_i_ and R_i_ Calculations from MD Simulation Data

The algorithm implementation for calculation of T_i_ and R_i_ from MD simulation showed to be reliable when reproducing experimental data. Both published works of Gonçalves et al. [45] and Lino et al. [47] used the algorithm explained in the previous section to compute NMR spin relaxation parameters from MD simulation data. Table 1 summarizes the results obtained for validation.

In fact, magnetite [45] can be used in order to validate the algorithm. In this case, it obtained an R_1_ value of 35.72 s^−1^, which was just 4.47 s^−1^ different from the experimental one, with an R_1_ value of 31.25 s^−1^. For R_2_, the obtained value was 55.55 s^−1^, only 5.05 s^−1^ different from the experimentally measured one, which was 50.50 s^−1^. In addition, R_1_/R_2_ ratios were very similar, being that the theoretically calculated equals 0.643 and the experimental one is equivalent to 0.619. In addition, from the work published by Lino et al. [47], trichloroethylene (TCE) and iodotrifluoroethylene (TFE) were also used to validate the algorithm. For the theoretical calculations for TCE, C-C distance was used, and for TFE, F-1-F-2 and F-2-F-3 distances were used. For TCE(C-C), the calculated T_1_ value was 8.98 s, very close to the experimental data value, which was 8.90 s. For TFE(3F-4F) and TFE(5F-3F) the T_1_ values were 5.35 and 5.52 s, respectively, also very close to the experimental data values, which were 5.37 and 5.56 s, respectively. The same is valid for T_2_, which is also shown to be very similar.

Vanadium complexes could be simulated in order to validate the algorithm, therefore, there is a lack of literature describing NMR relaxation parameters for vanadium complexes, especially experimental relaxation times and rates in order to make a comparison. Thus, considering the nature of the algorithm (sensibility to average distance variation) and used inputs (magnetic field, gyromagnetic radius, and spin), the use of other compounds to validate the algorithm and perform a theoretical study about NMR relaxation properties was necessary.

In this scenario, previous papers showed that the discussed algorithm can be a reliable tool for calculating NMR spin relaxation parameters. This way, the algorithm can be used for a theoretical study of VC1 and VC2 as potential MRI contrast agents for cancer and AD detection.

### 2.3. MD Simulation of VC1 and VC2 in Water Only and in the Presence of the Respective Protein Targets

The RMSD analyses for the VCs in each medium can be found in Figure 2, in which VC1 and VC2 showed stable conformations throughout the entire time of the simulation. It can be observed as well that the complexes reached equilibrium conditions in both WAT and protein + WAT environments.

Specifically, VC1 exhibited an average RMSD value of 0.3335 Å with a standard deviation of 0.0035 Å in the water surroundings, while an average RMSD value of 0.2421 Å with a standard deviation of 0.0013 Å was found for VC1 in the protein + WAT environment.

Similarly, VC2 displayed a slight difference in RMSD values. In the water medium, the average RMSD value was 0.3200 Å with a standard deviation of 0.0020 Å, whereas in the presence of the biological target, the average RMSD was equal to 0.3086 Å with a standard deviation of 0.0016 Å.

The difference between VC1 + WAT and VC1 + AMPK + WAT, as well as VC2 + WAT and VC2 + ULK1 + WAT, can be explained by the fact that when the respective proteins were present, the VCs had even more restricted movements than solely in water. As a result, the complexes exhibited reduced oscillation freedom.

### 2.4. OWSCA Conformational Selecting and Calculated T_i_ and R_i_ Values

From MD simulation data, the OWSCA methodology was applied in order to reduce the data for further T_i_ and R_i_ calculations. Figure 3 shows the selected conformations for all systems. Therefore, the OWSCA methodology was able to reduce MD simulation data of 2000 conformations for each system in about 100 conformations, varying between 96 and 119 conformations. It is also possible to see that the use of the db1 (haar) wavelet was able to capture the MD simulation data behavior, implying that the important information is contained in the compressed dataset.

Both the relaxation times (T_i_) and their respective relaxation rates (R_i_) were calculated using Equations (1)–(7) and presented in Table 2. It is important to notice that all values resided very close to each other, except the one corresponding to the system VC2 + ULK1 + WAT, which was considerably higher than others. In addition, Table 2 presents the R_1_/R_2_ ratio for all formulations. It is possible to see the similarity between them, including the magnetite system used for validation of the R_1_/R_2_ ratio, provided in Table 1, indicating the consistency of the calculation.

In order to understand the presented results, it is important to take a closer look at the systems, which have different characteristics in terms of structure, size, and ligands. Figure 4 shows the structure of both VCs studied. Both structures present a double bond between vanadium and oxygen atoms, and the V=O group is an important region in which a non-coordinated water molecule can interact with by a hydrogen bonding interaction [55]. Hence, most of the water molecules present in the chemical medium will approach more closely to the metal atom at the center of the complexes due to the presence of the V=O group since the resulting interaction is considerably strong. Therefore, the availability of this group for interactions results in closer distances between ^1^H and V(IV). The MD simulation of the built models containing only VC + WAT showed an average distance V⋯^1^H for VC1 of 3.19 Å, and for VC2, 3.66 Å.

When comparing the structures presented in Figure 4a,b, it is possible to observe that VC1 has a smaller coordinated group, and VC2 a larger coordinated group. This group is also more hydrophobic, due to the major presence of carbons instead of nitrogen atoms, causing the increased V⋯^1^H average distance in VC2 + WAT, as mentioned before. Therefore, the two characteristics mentioned imply considerable interaction differences between the VCs and the chemical medium.

The dynamics of interactions between VCs, water, and the target proteins play a relevant role in the observed results. As discussed previously, the main path in which a water molecule can approach the central metal atom is via the V=O group. Therefore, if this site of interaction is unavailable, it is expected that the water molecules will be farther apart from the vanadium atom.

Figure 5a shows the interaction between VC1 and AMPK. It is possible to see that the interaction occurs in a way that the V=O group becomes more unavailable for hydrogen bonding interaction with water. Despite that, there is still a possible way for the water to approach the vanadium atom, since the coordination groups are smaller and more hydrophilic. Hence, the average V⋯^1^H distance observed for the system VC1 + AMPK + WAT was 3.29 Å, only 0.10 Å higher than for the system VC1 + WAT.

On the other hand, Figure 5b shows the interaction between VC2 and ULK1. The interaction with the protein occurs mostly with the V=O group, making it completely unavailable for interactions with other molecules. The coordinated groups are bigger and more hydrophobic, leaving no considerable space for the water molecules to approach the vanadium atom, due to a relevant steric hindrance. Therefore, the average V⋯^1^H distance observed for the system VC2 + ULK1 + WAT was 5.55 Å, 1.89 Å higher than for the system VC2 + WAT.

Comparing both systems with their respective proteins, it is possible to notice that the V=O group of VC2 becomes completely unavailable when it interacts with ULK1, leading to a more difficult interaction with the water in the medium, increasing the average V⋯^1^H distance by a considerable amount. In addition to the fact that VC1 also interacts with AMPK by the V=O group, this site of interaction does not become completely unavailable, allowing the water molecules to approach the vanadium atom, which leads to a very small increase in the average V⋯^1^H distance.

From Equations (5)–(7), it is possible to notice how the average distances ϕ between the vanadium atom and hydrogen atom of the water molecule play an important role in the T_1_ and T_2_ calculations. An increase in the average distance also increases the relaxation time, decreasing the relaxation rate. Due to the high order power of ϕ, a small increase in its value has a considerable impact on T_i_ and R_i_ values.

This way, the mathematical relationship between ϕ and T_i_ can be interpreted based on relaxation concepts. Both longitudinal and transversal relaxation times are a measurement of how quickly a molecule can return to its equilibrium state after the removal of a radio pulse in the presence of a magnetic field B_0_. This way, the return to its equilibrium state is possible due to the energy distribution with the surroundings [56,57]. Hence, when evaluating the influence of VCs in ^1^H relaxation time, it is assumed that the VCs are important agents for receiving this dissipated energy, contributing to a faster returning of ^1^H to its equilibrium state. For that distribution to be more effective, it is important that the distance V⋯^1^H should be as close as possible. Therefore, in a situation where the distance between the vanadium atom and ^1^H is larger, it is expected to have a higher relaxation time.

From our findings regarding the VC2 system with ULK1, the unavailability of the V=O for hydrogen bonding interactions and also the size and affinity with water of the coordinated groups resulted in an increase in average V⋯^1^H distance. This increase resulted in a considerably higher T_i_ value when compared with the other systems.

From the obtained results, it was possible to make a comparison between the proposed potential contrast agents with those commercially available. For Gd-DOTA, a commercial Gd(III)-based contrast agent, its T_1_ and T_2_ have a value of 0.032 and 0.025 s, respectively [52,53]. For Gd-DTPA, also one of the most common contrast agents, the T_2_ value is 0.020 s [46,50,55]. Then, the VC1 showed to be a potential longitudinal and transversal MRI probe, being more effective than Gd-DOTA and Gd-DTPA, with superior relaxation times. In fact, its T_1_ and T_2_ values were 0.084 and 0.056 s, respectively, considering interactions in a chemical medium with only water, and 0.074 and 0.050 considering the presence of AMPK. Although VC2 presented a T_1_ and T_2_ of 0.086 and 0.058 s, respectively, when considering the system only with water, its effectiveness did not hold for a system considering interactions with ULK1, in which T_1_ and T_2_ were 1.046 and 0.702 s.

Despite the fact that VC2 did not show to be more effective than Gd-DOTA and Gd-DTPA when considering ULK1 as the protein target, its obtained values for T_i_ and R_i_ suggest that modifications in the structure of VC2 might lead to a more effective potential contrast agent. The foundation for this proposal remains in the fact that since the unavailability of the V=O group increases T_i_ value, an addition or substitution of a ligand, capable of interacting in the same ULK1 site as V=O, might make it more available for hydrogen bonding interactions, allowing the water molecules to approach the vanadium atom and dissipate energy faster.

## 3. Materials and Methods

### 3.1. Systems Descriptions and Docking Studies

In total, four systems were studied, being [VO(metf)_2_]·H_2_O (VC1) in a solvation box (VC1 + WAT), VC1 and its biological target AMPK with water molecules (VC1 + AMPK + WAT), [VO(bpy)_2_Cl]^+^ (VC2) with only water molecules (VC2 + WAT), and VC2 with its target protein in aqueous medium (VC2 + ULK1 + WAT). The chemical structures of the vanadium complexes are shown in Figure 6.

Information about systems VC1 + WAT and VC1 + AMPK + WAT is based on the paper published by Tavares, C. A. et al. [38], while information about systems VC2 + WAT and VC2 + ULK1 is based on the work of Santos, T. M. R. et al. [39]. The three-dimensional structure of the AMPK protein was obtained from the RCSB Protein Data Bank (PDB ID: 6C9G) [58]. Since some amino acid residues were absent from the obtained structure, the preparation of the protein was performed using the SWISS-MODEL platform [59]. For the three-dimensional structure of the ULK1 protein (PDB ID: 4WNO) [60], no other previous treatments were performed since the original file contained all necessary residues. For both biological targets, hydrogen atoms were added to the protein structures and charges were calculated by using BIOVIA Discovery Studio v.21 [61].

Docking studies were carried out by using the Molegro Virtual Docker [61,62] for both AMPK and ULK1 structures. The binding site chosen for the docking study associated to AMPK was based on the work by [63], where it was reported that the independent ligand of VC1 (metformin) interacted with residues Asp-215, Asp-216, and Asp-217 in the homology model. Flexible residues were included within a radius of 8 Å, and the binding site radius was set as 7 Å. In turn, for ULK1, the binding site considered in this study was derived from [64]. Moreover, flexible residues were set to be within a 12 Å radius.

### 3.2. Molecular Dynamics Simulations

The paper published by Tavares, C. A. et al. [38] developed AMBER force field parameters for VC1, which were validated in the same work. In another study, made by Santos, T. M. R. et al. [39], parameters for a new AMBER force field for VC2 were developed and subsequently validated. Therefore, the development of such parameters for both VC1 and VC2 allowed the investigation carried out in this paper, where the simulations were performed using the AMBER20 package.

All MD simulations were carried out with a total time of 200 ns, following the steps of minimization, heating, equilibration, and production. Once again, four systems were investigated through this Molecular Mechanics technique.

For AMPK, the ff99SB-ILDN force field was used to describe the protein. The first step was performed by minimizing the energy of the system using the steepest descent and, subsequently, the conjugate gradient method. In the heating step, the temperature was gradually increased from 0 to 300 K, then, the system was equilibrated at the same temperature, with a gradual decrease in the restriction. For the production step, an explicit solvent simulation was conducted using the TIP3P model for water molecules.

For ULK1, the ff19SB force field was used to simulate this system. The system containing this protein underwent the minimization step using the steepest descent followed by the conjugate gradient at constant volume. Next, the heating of VC2 + ULK1 + WAT was carried out, reaching 300 K, where a restriction of 500 kcal/mol was applied. Subsequently, the system was equilibrated at the same temperature of the last step. Lastly, the production step was performed using the OPC water model.

The MD simulations of VC1 + WAT and VC2 + WAT were carried out in a similar procedure, at 300 K using a solvation box with cubic dimensions and TIP3P and OPC water models to represent the water molecules for the respective systems.

### 3.3. Conformational Selections of V(IV) Complexes Using the Optimal Wavelet Signal Compression Algorithm and MD Data for T1 and R1 Estimation

The Optimal Wavelet Signal Compression Algorithm (OWSCA) showed to be a successful resource to treat different systems, especially when dealing with possible MRI probes [65]. In fact, it can be used to select the most representative conformations in the dataset, reducing the data to be applied in further calculations.

OWSCA is based on discrete wavelet transform of the dataset x(t), and it can be defined according to Equation (8), wherein such conversion reveals the main features of the system and, therefore, represents the original signal in a reduced way $\tilde{x}$(t), without losing important characteristics of the system [45,65].
(8)$$d_{j,k}=\int_{-\infty }^{+\infty }x\left(t\right)\psi_{j,k}\left(t\right)dt$$
where d_j,k_ is the wavelet coefficient, t is the time normalized between 0 and 1, j represents the scaling parameter responsible to determine the time and frequency resolutions of the scaled wavelet function ψ, and k represents the shifting parameter, which translates the scaled wavelet along the time axis [45]. The wavelet ψ has oscillating wave-like characteristics and has it concentrated in time or space. Consequently, there are several types and families of wavelets, whose properties differ along convergence speeds when time tends to 0, symmetry, compression potential, and smoothness [45]. Thus, when using OWSCA for reducing the original dataset and maintaining the principal characteristics of the system, the choice of the appropriate wavelet is an important step to consider.

With the MD simulation data for the four systems, VC1 + WAT (1), VC2 + WAT (2), VC1 + AMPK + WAT (3), and VC2 + ULK1 + WAT (4), the OWSCA was applied in order to reduce the data for further calculations. Considering the nature of the systems, it is important to keep in mind that the main contribution of the total energy of systems (3) and (4) is associated with the protein, and the main contribution of the total energy of systems (1) and (2) is due to the V(IV) complexes. These assumptions were made based on the fact that they are the large molecules of their respective system. The chosen wavelet for all systems was db1 (haar), which is capable of representing signals with non-smooth transitions, since it presents a discontinuous profile [66]. Then, T_i_ and R_i_, where *i* = 1 indicates longitudinal and *i* = 2 indicates transverse, were estimated from the computed V⋯^1^H distances, obtained for each conformation selected by the OWSCA procedure. The flowchart shown in Figure 7 summarizes the needed steps for the work purposes. The OWSCA procedure is implemented in a homemade software [67,68,69].

## 4. Conclusions

From the presented work, it was possible to conclude that vanadium complexes could be promising contrast agents. Another important conclusion made from this work, which reinforces the promising potential of vanadium complexes as MRI probes, is that both systems studied for VC1 showed to be more effective than Gd-DOTA and Gd-DTPA, two available commercial contrast agents for MRI. Despite the fact that VC2 did not show to be as good as Gd-DOTA and Gd-DTPA when considering the relaxation parameters with its target protein, ULK1, the presented results contribute to the development of less toxic and more efficient MRI probes. In addition, as described in previous works, vanadium complexes have a promising potential in therapy of many diseases, but there is a lack of literature describing its potential as contrast agents. In this scenario, acting as MRI probes, vanadium complexes could be used for theranostics of many types of cancer and Alzheimer’s, including the facilitation of vanadium-based drugs design. In light of the obtained results, it is possible to encourage new efforts on testing new VCs as MRI contrast agents, aspiring to a more effective diagnosis and possible new therapies.

Furthermore, the work presents a fresh perspective about how to assess VCs’ NMR relaxation properties. From the presented methodology, it was shown the possibility of theoretically studying of vanadium systems, including specific targets and considering explicit solvent effects. Therefore, it is possible to study complex biological systems, providing insights for further efforts and experimental studies.

Thus, the presented study is another step toward pursuing improvements in cancer and AD early diagnosis, also concerning safer contrast agents for patients, and giving a potential tool for new drug design. We hope, then, that our results will stimulate new experimental and full-dimensional theoretical investigations that could assess the validity of this assumption.

## Figures and Tables

**Figure 1 pharmaceuticals-16-01653-f001:**
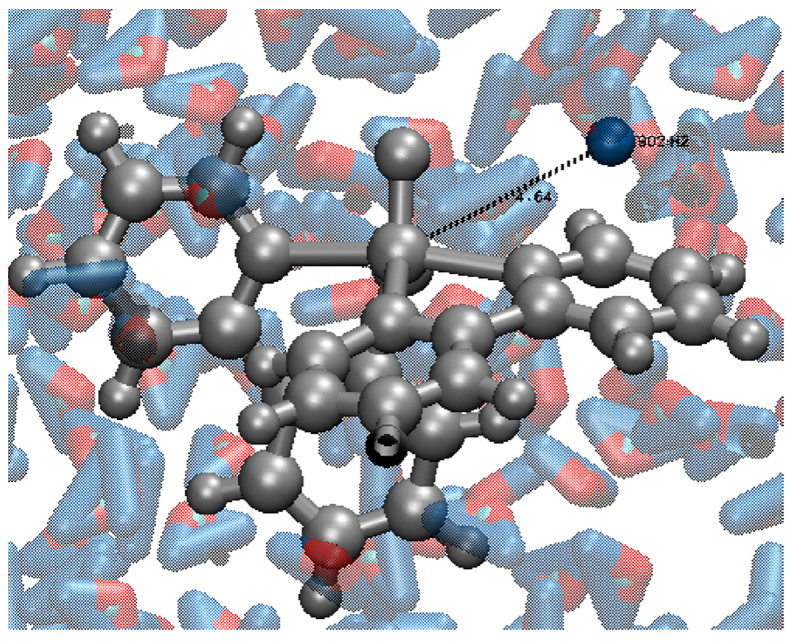
Representation of the measured V⋯^1^H distances. The central gray sphere represents the vanadium atom, and the blue sphere represents the ^1^H proton of the closest water molecule, being both connected by a black dashed line, representing the measured distance.

**Figure 2 pharmaceuticals-16-01653-f002:**
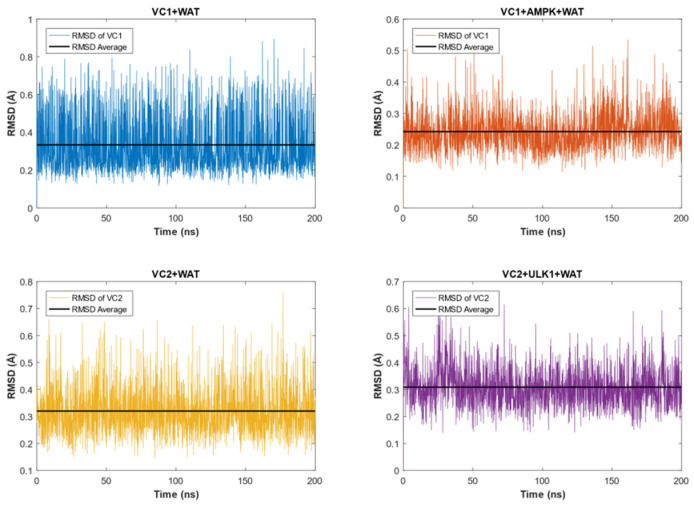
RMSD analyses of each system under investigation. The results are as follows: VC1 in the presence of WAT (depicted in blue), VC1 in the presence of AMPK and WAT (depicted in orange), VC2 in WAT (depicted in yellow), and VC2 in the presence of ULK1 and WAT molecules (depicted in purple).

**Figure 3 pharmaceuticals-16-01653-f003:**
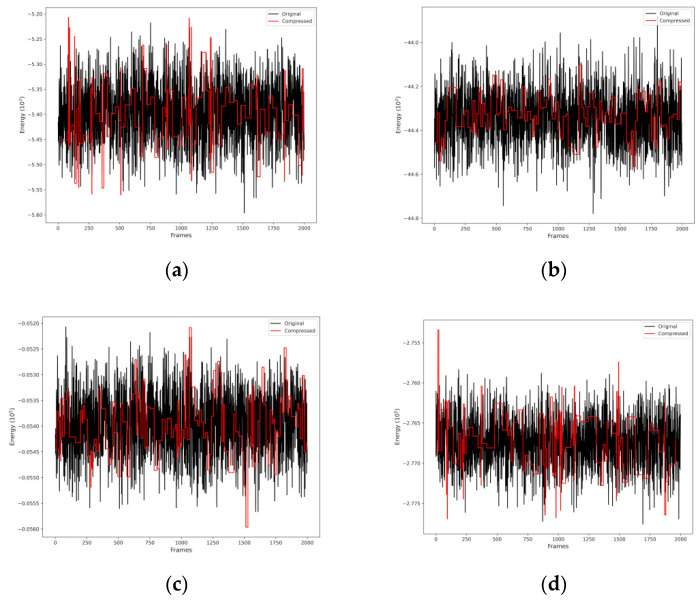
OWSCA conformational selection for (**a**) VC1 + WAT, (**b**) VC2 + WAT, (**c**) VC1 + AMPK + WAT, and (**d**) VC2 + ULK1 + WAT.

**Figure 4 pharmaceuticals-16-01653-f004:**
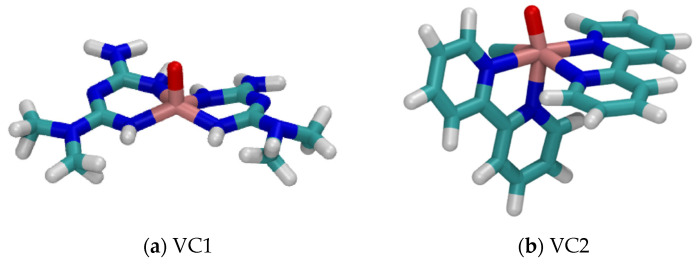
Structures of the V(IV) complexes in study, where the red region denotes the double bond between V and O.

**Figure 5 pharmaceuticals-16-01653-f005:**
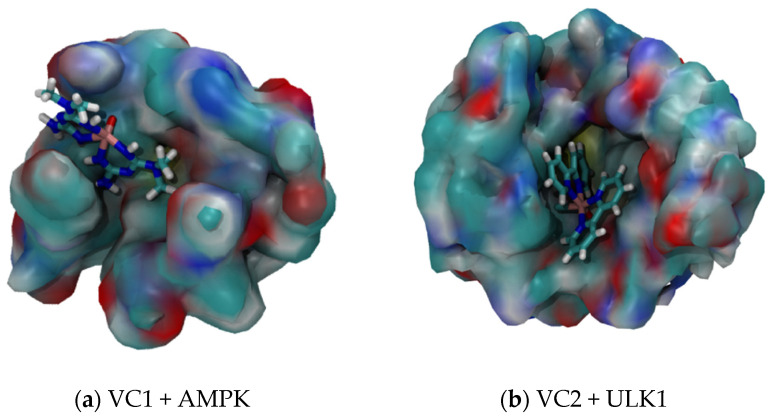
Interaction between vanadium complexes and their respective proteins of the system.

**Figure 6 pharmaceuticals-16-01653-f006:**
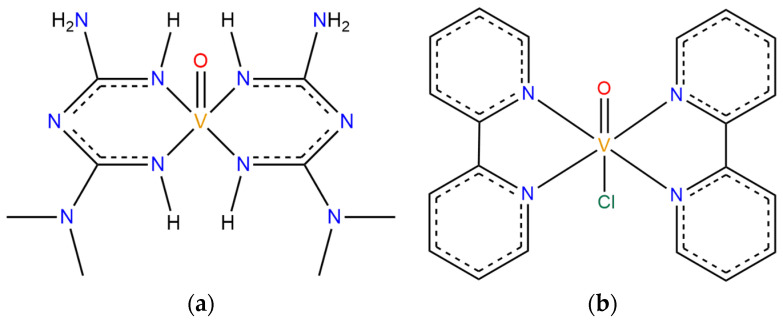
Chemical structures of (**a**) VC1: [VO(metf)_2_]·H_2_O and (**b**) VC2: [VO(bpy)_2_Cl]^+^.

**Figure 7 pharmaceuticals-16-01653-f007:**
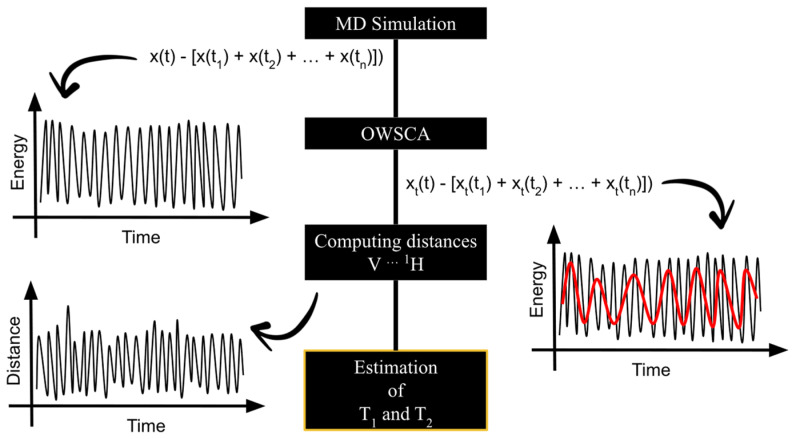
Flowchart of the steps followed for the estimation of T_1_ and T_2_ from MD simulation data.

**Table 1 pharmaceuticals-16-01653-t001:** R_i_ and T_i_ values for the validation systems in s^−1^ and s, respectively.

System	Theoretical				Experimental			
	T_1_	R_1_	T_2_	R_2_	T_1_	R_1_	T_2_	R_2_
Magnetite	0.028	35.72	0.018	55.55	0.032	31.25 [48]	0.020	50.50 [49,50]
TCE(C-C)	8.98	0.11	1.17	0.85	8.90 [51]	0.11	1.18 [51]	0.85
TFE(3F-4F)	5.35	0.18	0.12	8.33	5.37 [51]	0.19	0.14 [51]	7.14
TFE(5F-3F)	5.52	0.10	0.10	10.00	5.56 [51]	0.18	0.12 [51]	8.33

**Table 2 pharmaceuticals-16-01653-t002:** R_i_ and T_i_ values for all V(IV) complex systems in s^−1^ and s, respectively. In addition, R_1_/R_2_ values were calculated for all the formulations.

System	T_1_	T_2_	R_1_	R_2_	R_1_/R_2_
VC1 + WAT	0.084	0.056	11.905	17.921	0.664
VC1 + AMPK + WAT	0.074	0.050	13.514	20.121	0.672
VC2 + WAT	0.086	0.058	11.630	17.331	0.671
VC2 + ULK1 + WAT	1.046	0.702	0.956	1.424	0.671
*Gd-DOTA (DOTAREM)*	0.032 [52]	0.025 [52,53]	31.250	40.000	0.781
*Gd-DTPA (MAGNEVIST)*	-	0.020 [46,50,54]	-	50.000	-

## Data Availability

Data is contained within the article.

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
