# Peer review of "MD Simulations to Calculate NMR Relaxation Parameters of Vanadium(IV) Complexes: A Promising Diagnostic Tool for Cancer and Alzheimer’s Disease"

_pharmaceuticals, 2023, doi:10.3390/ph16121653_

Round 1
Reviewer 1 Report
Comments and Suggestions for Authors
Rodrigo Mancini Santos and colleagues have submitted a study of the behavior of two oxidovanadium (IV) complexes in selected protein targets and aqueous environment by molecular dynamics simulations.
The submitted manuscript overall demonstrates competent execution and explanation of experiments and results. However, in the subsequent section, I will provide a thorough analysis, addressing both minor and major aspects of the manuscript that require attention to enhance its quality before it can be considered for publication in Pharmaceutics.
Minor aspects:
Abstract
- Correct the 2 of the crystallization water (subscript)
- Metf and bpy are not defined in the abstract
Introduction
- Please name the compounds adding square brackets in the coordination compound: e. g. [VO(metf)2]·H2O
- Line 36: Correct “personalized”
- Line 56: Aclare “Vanadium(IV)”, not only vanadium. Because this is the paramagnetic ion that it is mentioned in the following line. Also, delete “usually”, because vanadium(IV) is always paramagnetic.
- Line 59: Vanadium complexes are labile. They are not stable, in coordination chemistry terms, but also redox terms. Please delete the sentence “excellent stability”.
- Line 61: Define “common concentrations”. Put a range or refer something as Redher writes in the referenced review.
- Line 70: Cite also: Pessoa, J. C., Etcheverry, S., & Gambino, D. (2015). Vanadium compounds in medicine. Coord Chem Rev, 301, 24-48. doi:10.1016/j.ccr.2014.12.002
- Line 74: Include a sentence, “where metf is …name…. and bpy is …. Name…”
- Line 82: define the acronymus ULK1
Methods:
- As far as I know, Pharmaceutics organize the papers first the results and after the Methods. Please organize the manuscript according to the Journal specifications.
Major aspects:
Particularly in Figure 1, but this comment works for the whole manuscript: The compound VO(bpy)2Cl have a mistake in the charge. If Vanadium is IV, it should be a cation complex. Please provide the right information about the chemistry and charge of the molecule. Write the formula in the wrong way is unacceptable.
Author Response
Dear Editor,
Pharmaceuticals
We very much appreciate the response e-mail containing the valuable and helpful reviewer evaluation of our manuscript. Please, find enclosed the new version including the suggestions made by Reviewers 1 and 2.
Accordingly, we carried out a complete revision of the manuscript with emphasis on the scientific content and on the formatting request. We hope that all the necessary improvements of the discussion were implemented. The list of changes and responses to the referees’ comments are listed below and modified parts of the paper are highlighted in yellow.
Reviewer Comments:
Reviewer 1:
Minor aspects:
Abstract
- Correct the 2 of the crystallization water (subscript).
Author reply: We would like to thank you for your observation. The formatting has been corrected as necessary and can be checked in line 19.
- Metf and bpy are not defined in the abstract.
Author reply: Thank you for the observation. We agree with the referee’s comment and understand the importance of defining such acronyms in the abstract. Therefore, both acronyms were defined in the abstract, as can be seen in lines 20-21.
Introduction
- Please name the compounds adding square brackets in the coordination compound: e. g. [VO(metf)2]·H2O.
Author reply: Thank you for this comment, we believe this is an important observation, we appreciate it. In this scenario, square brackets were added in the coordination compound notations all along the manuscript.
- Line 36: Correct “personalized”.
Author reply: We thank you for the observation. It has been corrected as noted by the reviewer.
- Line 56: Aclare “Vanadium(IV)”, not only vanadium. Because this is the paramagnetic ion that it is mentioned in the following line. Also, delete “usually”, because vanadium(IV) is always paramagnetic.
Author reply: Thank you for this comment, we believe this is a great observation, we appreciate it. Thus, we specified in line 56 (now 73) that we are talking about vanadium(IV) complexes, and also removed the word “usually”, once the correct sentence is that vanadium(IV) is always paramagnetic.
- Line 59: Vanadium complexes are labile. They are not stable, in coordination chemistry terms, but also redox terms. Please delete the sentence “excellent stability”.
Author reply: Thank you for this useful perspective. We would like to say that we removed the sentence “excellent stability” in line 59 (now 75), once it is technically incorrect in coordination chemistry and redox terms.
- Line 61: Define “common concentrations”. Put a range or refer something as Redher writes in the referenced review.
Author reply: Thank you for the observation. In line with the referee's comment, we have changed this sentence in order to make it clear to the reader information about vanadium toxicity. The new sentence can be seen in lines 77-78.
- Line 70: Cite also: Pessoa, J. C., Etcheverry, S., & Gambino, D. (2015). Vanadium compounds in medicine. Coord Chem Rev, 301, 24-48. doi:10.1016/j.ccr.2014.12.002.
Author reply: We appreciate the suggestion. Also, we think that the inclusion of this reference is of great importance. Therefore, the suggested work was cited, as can be seen in line 89.
- Line 74: Include a sentence, “where metf is …name…. and bpy is …. Name…”.
Author reply: Your observation is appreciated. Now we defined the acronyms “metf” and “bpy”, and it can be seen in lines 93-94.
- Line 82: define the acronymus ULK1.
Author reply: We really appreciate this observation. Now we defined the acronym ULK1, which is Unc-51-like kinase 1, and it can be seen in line 101.
Methods
- As far as I know, Pharmaceutics organize the papers first the results and after the Methods. Please organize the manuscript according to the Journal specifications.
Author reply: Great observation, we appreciate it. Hence, we have changed the manuscript in order that the results came before the Methods, being in agreement with the Journal specifications.
Major aspects:
- Particularly in Figure 1, but this comment works for the whole manuscript: The compound VO(bpy)2Cl have a mistake in the charge. If Vanadium is IV, it should be a cation complex. Please provide the right information about the chemistry and charge of the molecule. Write the formula in the wrong way is unacceptable.
Author reply: Thank you for your concern. This comment is of great importance for us and our manuscript. We apologize for our mistake, unfortunately this writing error went unnoticed by us, and we agree with the referee’s comment that this is unacceptable. In this way, we emphasize that the writing of both vanadium(IV) complexes was carefully checked and corrected every time it appears in the manuscript. Additionally, we would like to clarify for the reviewer that the charges were accurately taken into account for every theoretical calculation; the mistake was solely in the written presentation. Once again, we apologize for that oversight and appreciate the important observation. Thank you for this important observation.
We acknowledge again the referee’s comments, which have enabled us to significantly improve our paper.
Furthermore, we thank the editorial assistance and hope with the changes and clarifications implemented, the manuscript would be now acceptable for publication in Pharmaceuticals. Finally, we also remain at your disposal for any further inquiries.
With best regards,
Teodorico C. Ramalho
Department of Chemistry,
Federal University of Lavras,
Brazil.
e-mail: teo@ufla.br
http://www.nucleoestudo.ufla.br/gqc/
https://orcid.org/0000-0002-7324-1353

Reviewer 2 Report
Comments and Suggestions for Authors
The manuscript entitled “MD Simulations to Calculate NMR relaxation parameters of Vanadium (IV) Complexes: A Promising Diagnostic Tool for Cancer and Alzheimer’s Disease” written by Rodrigo Mancini Santos et al, aims to investigate the interaction between two known vanadium complexes, VO(metf)2·H2O (VC1) and VO(bpy)2Cl (VC2), and the biological targets AMPK and ULK1, biomolecules related to Alzheimer's disease and triple-negative breast cancer. To this purpose, the authors evaluate the interaction between the vanadium complexes and the potential ligands AMPK and ULK1 in protein and aqueous environment by performing four classical molecular dynamics (MD) simulations. By using the Optimal Wavelet Signal Compression Algorithm (OWSCA) method the author calculated the relaxation parameters T1 and T2 which permitted them to highlight the versatility of vanadium complexes in the diagnosis and treatment of different diseases, like cancer or Alzheimer's disease.
1. Overall, the manuscript is well written and has a good scientific impact on highlighting new promising drugs for diagnosis and treatment.
2. The authors did not cite in the Introduction section some recent published articles. Please take into account the below-mentioned works:
1. Inorganics 2022, 10(4), 47; https://doi.org/10.3390/inorganics10040047
2. Biomedicines 2022 May 24;10(6):1217.
3. Biomedicines 2021 May 17;9(5):562.
4. Science China Life Sciences volume 62, pages126–139 (2019)
3. Why didn't the authors try to include in their study other existing vanadium complexes or some synthesized by them?
4. Why didn't the authors try to include in their study more specific biological targets for AD, like amyloid beta (Aβ) peptide, tau protein, apolipoprotein E4, etc?
5. The methodology is well-detailed and convincing.
6. More discussions comparing the authors' results with other already known data would strengthen the impact of the results of this manuscript.
7. The conclusions are a little ambiguous and leave the impression that the authors' results are not convincing and the essential idea is lost in the details. Also, the formulation of a general conclusion of the entire study would be impactful for the article.
Author Response
Dear Editor,
Pharmaceuticals
We very much appreciate the response e-mail containing the valuable and helpful reviewer evaluation of our manuscript. Please, find enclosed the new version including the suggestions made by Reviewers 1 and 2.
Accordingly, we carried out a complete revision of the manuscript with emphasis on the scientific content and on the formatting request. We hope that all the necessary improvements of the discussion were implemented. The list of changes and responses to the referees’ comments are listed below and modified parts of the paper are highlighted in yellow.
Reviewer Comments:
Reviewer 2:
- Overall, the manuscript is well written and has a good scientific impact on highlighting new promising drugs for diagnosis and treatment.
Author reply: We thank you for your commentary. We hope that our work may encourage future works on vanadium complexes as possible new drugs for treatment and new compounds for the diagnosis.
- The authors did not cite in the Introduction section some recent published articles. Please take into account the below-mentioned works:
- Inorganics 2022, 10(4), 47; https://doi.org/10.3390/inorganics10040047
- Biomedicines 2022 May 24;10(6):1217.
- Biomedicines 2021 May 17;9(5):562.
- Science China Life Sciences volume 62, pages 126–139 (2019)
Author reply: Thank you for this comment; we believe this is an important suggestion. We therefore added a sentence in the introduction section in order to emphasize the versatility and potential of vanadium complexes in the treatment of AD, cancer and diabetes in our work. In this sentence, the suggested works were cited, as can be seen in lines 84-86, page 2.
- Why didn't the authors try to include in their study other existing vanadium complexes or some synthesized by them?
Author reply: Thank you for this comment, we believe this is a great observation and appreciate it. The present work was executed with an entirely theoretical methodology, performing molecular dynamics simulations, and with the obtained data we have applied the OWSCA method in order to select conformations for obtaining NMR relaxation parameters from spectral density based calculations. Hence, this method is based on the previous studies (DOI: 10.1021/acs.jpcb.2c07147 and DOI: 10.1080/07391102.2023.2250453), where force fields were parameterized for both vanadium complexes investigated as promising new contrast agents for MRI with the aim of detecting Cancer and Alzheimer’s Disease. Therefore, the inclusion of new vanadium complexes requires the parametrization of new force fields in order to perform reliable molecular dynamics simulations, and further apply the data in the presented methodology.
Finally, our group does not have access to the synthesized vanadium complexes at this moment in order to make experimental measurements. However, we hope that in the near future we will be able to synthesize not only the two mentioned vanadium complexes, but others to further explore promising vanadium-based drugs for the diagnosis of Cancer and Alzheimer’s Disease.
- Why didn't the authors try to include in their study more specific biological targets for AD, like amyloid beta (Aβ) peptide, tau protein, apolipoprotein E4, etc?
Author reply: We thank you for the observation. We believe it is of great importance. As mentioned before, the investigation of the two vanadium complexes presented in the study as promising contrast agents for MRI was only possible due to previous force field parameterization, which allowed us to apply the described methodology. In this context, the study of various vanadium complexes, for which specific literature exists detailing their interaction with the mentioned targets, as noted by the reviewer, would necessitate the parametrization of new force fields. .
Hence, focusing specifically on [VO(metf)2]•H2O (VC1) and its target protein (AMPK), chosen for studying the potential of VC1 in the diagnosis of AD, previous studies have indicated an association between the development of AD and type 2 diabetes mellitus (T2DM). The link between these two conditions is the protein kinase AMPK, and the progression of T2DM has been suggested to lead to cognitive impairment. (DOI: 10.1016/j.bbr.2020.113043). Furthermore, there is a possibility that T2DM may affect the development of AD through insulin signaling in the brain (DOI: 10.3389/fgene.2022.1019860). Moreover, in AD, AMPK can reduce Aβ expression and indirectly inhibit hyperphosphorylation of tau, and in this sense, AMPK may be an excellent biological target in studies where the prevention and treatment of AD are the main goals (DOI: 10.1016/j.biochi.2021.11.008 ). Therefore, VC1 has shown to be efficient in the treatment of T2DM and is considered a potential agent against AD, justifying then why to study the interaction between VC1 and AMPK (DOI: 10.3390/ph15040453 ).
- The methodology is well-detailed and convincing.
Author reply: We thank you for this commentary. We really appreciate it.
- More discussions comparing the authors' results with other already known data would strengthen the impact of the results of this manuscript.
Author reply: Thank you for this useful perspective. We agree that presenting other comparisons would strengthen the impact of our results. Hence, we have added Gd-DTPA (MAGNEVIST) data to Table 2 (page 7), also adding it in the discussions, located between lines 322-334 (page 9). As it is possible to see, we only added the T2 value with its R2 value. Unfortunately, there is a lack of literature presenting experimental NMR relaxation data to these compounds, since there are many difficulties associated with these measurements.
- The conclusions are a little ambiguous and leave the impression that the authors' results are not convincing and the essential idea is lost in the details. Also, the formulation of a general conclusion of the entire study would be impactful for the article.
Author reply: Thank you for this comment, we believe this is a great observation and can enrich our contribution, we appreciate it. In this sense, we have modified the Conclusion section in order to provide more general conclusions, focusing on the novelty of the presented work, and emphasizing that the work provided two promising new contrast agents for the early diagnosis of cancer and AD with MRI examination.
We acknowledge again the referee’s comments, which have enabled us to significantly improve our paper.
Furthermore, we thank the editorial assistance and hope with the changes and clarifications implemented, the manuscript would be now acceptable for publication in Pharmaceuticals. Finally, we also remain at your disposal for any further inquiries.
With best regards,
Teodorico C. Ramalho
Department of Chemistry,
Federal University of Lavras,
Campus Universitário, C.P. 3037, 37200-000, Lavras,
Brazil.
e-mail: teo@ufla.br
http://www.nucleoestudo.ufla.br/gqc/
https://orcid.org/0000-0002-7324-1353
